# No evidence for population-level benefits of polyandry in sharks and rays

**Joel H. Gayford**[1,2]*, **Estefanía M. Flores-Flores**[2]

**1** Department of Life Sciences, Imperial College London, London, United Kingdom, **2** Shark Measurements, London, United Kingdom

* jhg19@ic.ac.uk

**Data Availability Statement:** All relevant data are within the manuscript and its Supporting Information files.

**Funding:** The author(s) received no specific funding for this work.

## Abstract

Mating system variation refers to the spectrum between genetic monogamy and polyandry, and has important consequences for sexual conflict, sexual selection and individual fitness in animals. Theoretically this variation could also have substantial population-level effects, influencing population viability and extinction risk. Evidence for these effects is mixed, in part due to the fact that substantial environmental change is thought to be required for them to have visible demographic consequences. In this study we test for the presence of relationships between polyandry and population status in Elasmobranchii (sharks and rays). Elasmobranchii is a large vertebrate clade that exhibits substantial interspecific variation in both genetic mating system and population status, as well as being subject to intense anthropogenically-mediated environmental change. We also predict past macroevolutionary shifts in genetic mating system through elasmobranch phylogeny. Our results show that both genetic monogamy and polyandry have evolved multiple times independently within Elasmobranchii, and we suggest that both of these extremes represent alternative adaptive strategies that are favoured under discrete ecological and biological conditions. Nevertheless, there is no evidence of population-level consequences of mating system variation in elasmobranchs. These results are significant as they suggest that mating system variation in this clade is unlikely to be a major determinant of extinction vulnerability. Ultimately additional work will be required, however this study improves our understanding of the evolutionary dynamics underlying mating system variation in elasmobranchs, and the potential for resultant population-level consequences.

## Introduction

A wide spectrum of genetic mating systems is observed in the sexually reproducing organisms, ranging from strict genetic monogamy (where only one male contributes genetic material to a brood) to widespread polyandry [1–3]. Polyandry refers to the scenario in which multiple males contribute genetic material to a brood and is observed to varying extents in most major vertebrate radiations [2, 4–6]. Whilst the genetic mating systems of diverse taxa have been studied in detail, there remains substantial controversy regarding both the causes and consequences of variation in vertebrate mating systems [7–9]. The elevated fitness cost of sexual reproduction

**Competing interests:** The authors have declared that no competing interests exist.

for females, resulting from increased sexual conflict and selection for reduced male care is broadly accepted (reviewed in [10]). However, increasingly studies suggest that polyandry may convey direct or indirect fitness benefits to females, and the conditions under which polyandry should incur net fitness costs or benefits to females remain poorly understood [10, 11]. Hypothesised benefits of polyandry to females include reduced risk of genetic incompatibility from selfish genetic elements, increased fecundity, increased offspring genetic variation (and hence resilience to environmental change) and increased offspring viability [11].

The hypothesised consequences of monogamy and polyandry extend beyond individual fitness to population viability and extinction risk [10]. This is because polyandry results in greater sexual conflict between males and females [10, 11]. Under different conditions this elevated sexual conflict can either weaken or enhance sexual selection, with consequences for mutation load, the rate of adaptive evolution and the balance between natural selection and genetic drift, all of which are important determinants of extinction risk [10]. Consequently, there may be no universally superior genetic mating system, with polyandry and monogamy potentiall representing alternative adaptive strategies that are beneficial in different ecological contexts [12–14]. Presently, demographic consequences of mating system variation have been difficult to study as their detection requires some degree of environmental change that has a strong effect on birth and deaths rates within the population in question [10]. Consequently, improving our understanding of genetic mating system variation is not only of great eco-evolutionary interest, but provides material benefit from a conservation and management perspective.

Elasmobranchii (sharks and rays) is a large, ancient vertebrate radiation renowned for its high diversity of reproductive modes [15, 16]. Whilst most vertebrate clades exhibit either oviparity or placental viviparity, both of these strategies are abundant within Elasmobranchii, as well as various other forms of viviparity including histrotrophy and yolk-based viviparity [15, 17]. This diversity of reproductive modes has important consequences for sexual selection and sexual conflict, which in elasmobranchs can manifest as phenomena such as oophagy, coercive mating and intrauterine cannibalism [18–20]. Both genetic monogamy and polyandry have been reported in a number of elasmobranch taxa [21, 22], stimulating debate as to the underlying evolutionary mechanisms [21, 22]. Convenience polyandry (where the costs of resistance for females outweigh the costs of mating) is often assumed to underlie polyandry in this clade [22], whereas recent studies suggest that additional factors such as female choice and male-male competition are important [21]. However, to date only two studies have investigated polyandry in elasmobranchs in an evolutionary context [4, 23], modelling mating system variation as a single binary trait (multiple paternity or monogomy) which doesn't represent the full extent of variation observed in nature. Moreover, the potential population-level consequences of mating system variation in elasmobranchs remains entirely unstudied, which is notable given the concerning conservation status of many shark and ray species [24, 25].

In this study, we utilise data drawn from the literature to investigate the evolutionary history of polyandry in elasmobranchs, and test for signals of potential population-level consequences of mating system variation in a phylogenetic context. Elasmobranchii represents an ideal case study in this context given the extent of variation in mating systems that is observed in extant taxa (representing a spectrum between strict genetic monogamy and polyandry). Thus our results will not only provide additional context regarding the origins, causes and consequences of polyandry and genetic monogamy in sharks and rays, but improve our understanding of mating system variation more broadly. This context is of critical importance given the uncertainty regarding relationships between genetic mating system and extinction risk, and the impending anthropogenic extinction crisis.

## Methodology

### Ethics statement

No experimental work of any kind, working with either live or dead animals was carried out during this study, and consequently no ethics approval was required.

### Data collection

We collected data regarding polyandry (the frequency of multiple paternity, henceforth MP frequency) from existing literature, resulting in a dataset of 40 species (Table 1). MP frequency was defined as the percentage of litters fathered by more than one sire [2]. Incorporating a quantitative value of MP is hugely beneficial, particularly in evolutionary analyses, as it allows for species that exhibit intermediate mating systems and is less sensitive to error resulting from low sample sizes.

Where multiple studies have investigated polyandry in a single species (e.g. *Mustelus henlei*), MP frequency was averaged and weighted by sample size to provide a single measure per taxon. It is important to recognise that MP values could differ substantially between different populations of the same species, and thus averaged values may not be representative of the true genetic mating system of a given population. However in light of the available data, and the fact that our chosen proxies for extinction risk are also averaged across populations, we assert that this approach is suitable for making interspecific comparisons.

Quantifying individual evolutionary fitness or lifetime reproductive success in elasmobranchs (let alone averaging across populations) is notoriously challenging due to their size, life history parameters and ecology [26, 27]. For this reason, we collated readily available ecological/biological data that could serve as proxies for the average population status of elasmobranch species. Specifically we collected the following information: minimum and maximum litter size (Sharks of the World: a Complete Guide [28]), conservation status (IUCN [29]), 'lifetime reproductive output' [21], generation time (FishBase [30]), average fecundity (FishBase [30]), vulnerability to fishing (FishBase [30]), vulnerability to climate (FishBase [30]). If polyandry does indeed convey significant population-level benefits in elasmobranchs, we might (see discussion for caveats) expect these parameters to vary systematically with MP frequency. All data can be found in the supplementary materials associated with this article.

Phylogenetic data (a set of 10,000 phylogenies including branch lengths and interrelationships) were extracted from Stein et al. [25], and a pruned, maximum clade credibility (MCC) phylogeny was produced to match our dataset using the packages picante and phangorn [38, 39].

### Data analysis

All data analysis was carried out in R4.0.4 [40]. Prior to data analysis all quantitative variables were scaled.

To provide insight into the evolutionary dynamics underlying interspecific variation in MP frequency, we fit 3 evolutionary models to our data using the default parameters of the function fitContinuous in the package geiger [41]. Each of these models assumed a different model of trait evolution (Brownian motion, Ornstein Uhlenbeck and Early burst), and the model of best fit was selected on the basis of AIC values, where $\Delta AIC > 2$ indicates significant model support.

To test for potential population-level benefits of polyandry, we fit a series of phylogenetic generalised linear models using the package phylolm [42], with MP frequency as the predictor variable. These models can incorporate both continuous and discrete variables, and account for phylogenetic non-independence between the data [42]. Each model included one potential

**Table 1. MP frequency and presence/absence of polyandry in extant elasmobranch species.**

| Species | Genetic polyandry? | MP frequency (%; averaged) | Data source |
|---|---|---|---|
| *Raja clavata* | Yes | 100 | Lamarca et al., 2020 [4] |
| *Pristis pectinata* | No | 0 | Lamarca et al., 2020 [4] |
| *Urobatis halleri* | Yes | 90 | Lamarca et al., 2020 [4] |
| *Potamotrygon leopoldi* | Yes | 49.7 | Torres et al., 2022 [31] |
| *Aetobatus narinari* | Yes | 100 | Lamarca et al., 2020 [4] |
| *Hexanchus griseus* | Yes | 100 | Lamarca et al., 2020 [4] |
| *Pristiophorus nudipinnis* | Yes | 100 | Nevatte et al., 2023 [32] |
| *Pristiophorus cirratus* | Yes | 100 | Nevatte et al., 2023 [32] |
| *Squalus mitsukurii* | Yes | 11.1 | Lamarca et al., 2020 [4] |
| *Squalus acanthias* | Yes | 20.5 | Lamarca et al., 2020 [4] |
| *Etmopterus molleri* | No | 0 | Lamarca et al., 2020 [4] |
| *Etmopterus spinax* | Yes | 6.5 | Lamarca et al., 2020 [4] |
| *Rhincodon typus* | No | 0 | Lamarca et al., 2020 [4] |
| *Ginglymostoma cirratum* | Yes | 66.7 | Lamarca et al., 2020 [4] |
| *Scyliorhinus canicula* | Yes | 92.3 | Lamarca et al., 2020 [4] |
| *Galeocerdo cuvier* | No | 0 | Lamarca et al., 2020 [4] |
| *Sphyrna lewini* | Yes | 61.1 | Lamarca et al. 2020 [4] |
| *Sphyrna tiburo* | Yes | 18.8 | Lamarca et al. 2020 [4] |
| *Carcharhinus altimus* | Yes | 100 | Lamarca et al. 2020 [4] |
| *Carcharhinus plumbeus* | Yes | 65.1 | Lamarca et al. 2020 [4] |
| *Carcharhinus leucas* | Yes | 67.4 | Lamarca et al. 2020 [4] |
| *Carcharhinus acronotus* | Yes | 74 | Lamarca et al. 2020 [4] |
| *Carcharhinus isodon* | Yes | 83.8 | Nash et al. 2021 [33] |
| *Carcharhinus amblyrhynchos* | Yes | 66.7 | Lamarca et al. 2020 [4] |
| *Prionace glauca* | Yes | 91.5 | Armada-Tapia et al. 2023 [34] |
| *Carcharhinus obscurus* | Yes | 35.7 | Lamarca et al. 2020 [4] |
| *Carcharhinus galapagensis* | No | 0 | Lamarca et al. 2020 [4] |
| *Negaprion acutidens* | Yes | 77.8 | Lamarca et al. 2020 [4] |
| *Negaprion brevirostris* | Yes | 84.4 | Lamarca et al. 2020 [4] |
| *Galeorhinus galeus* | Yes | 40 | Lamarca et al. 2020 [4] |
| *Triakis semifasciata* | Yes | 36.4 | Lamarca et al. 2020 [4] |
| *Mustelus henlei* | Yes | 63.8 | Lamarca et al. 2020; Rendón-Herrera et al. 2022 [4, 35] |
| *Mustelus californicus* | No | 0 | Tárula-Marín and Saavedra-Sotelo. 2021 [36] |
| *Mustelus mustelus* | Yes | 50 | Lamarca et al. 2020 [4] |
| *Mustelus punctulatus* | Yes | 53.8 | Lamarca et al. 2020 [4] |
| *Mustelus asterias* | Yes | 58.3 | Lamarca et al. 2020 [4] |
| *Mustelus antarcticus* | Yes | 24.1 | Lamarca et al. 2020 [4] |
| *Mustelus lenticulatus* | Yes | 15.8 | Lamarca et al. 2020 [4] |
| *Carcharias taurus* | Yes | 42.9 | Lamarca et al. 2020 [4] |
| *Isurus oxyrinchus* | Yes | 83.3 | Liu et al. 2020 [37] |

covariate, and assumed the model of trait evolution best supported by the previous analysis (see results for details). Importantly not all covariate data could be collected for all species, and thus each of these models differs in sample size and cannot be directly compared on the basis of AIC values.

Finally, to estimate evolutionary shifts in MP frequency through elasmobranch phylogeny, we performed ancestral state reconstruction of MP frequency using the package phytools [43].

## Results

Comparison of Brownian motion (BM), Ornstein Uhlenbeck (OU) and Early burst (EB) models of trait evolution revealed that the phylogenetic distribution of MP frequency values is best explained by an OU model (Table 2). The EB model was the worst-performing of the three models (Table 2).

None of the nine phylogenetic linear models fitted produced evidence of significant correlation between MP frequency and biological/ecological proxies of population status (Table 3).

Ancestral state reconstruction of MP frequency suggests that basal elasmobranchs displayed intermediate levels of genetic polyandry (MP frequency at ancestral node was 59.6%), although support for this result was extremely weak and the confidence interval overlapped both 0 and 100. This dataset is likely of insufficient size to provide a robust estimate of the ancestral state. Nevertheless, this analysis did provide valuable insight into subsequent evolutionary shifts in polyandry, suggesting that both extremely high and low MP frequency values have evolved multiple times independently (Fig 1). There appear to be at least six independent transitions from polyandry to genetic monogamy, and even more cases of high MP frequency evolving from ancestrally intermediate MP frequency values (Fig 1).

## Discussion

### Population-level consequences of polyandry

Despite numerous predicted consequences for population fitness, we found no association between the extent of polyandry (MP frequency) and various measures of population status (Table 3). Assuming a positive relationship between polyandry and the strength of sexual selection (a link that is still somewhat controversial), polyandry could prove beneficial either by increasing genetic variation or by reducing the reproductive success of 'inferior' males, thus purging less favourable mutations from the population over time [10, 44, 45]. Moreover, if sexual selection happens to favour similar trait optima to natural selection, polyandry could enhance the rate of adaptive evolution [10]. However, the potential for sexual selection to oppose rather than enhance natural selection is well known [46, 47], and thus depending on the adaptive landscape of the male and female traits in question, polyandry could either increase or retard the pace of adaptation. Whilst there is some empirical evidence for population-level consequences of polyandry, most of these putative links do not have visible demographic consequences, as to have noticeable impact on genetic diversity or population size they require birth/death rates to be modified by environmental change [10, 48, 49].

This should provide no barrier to the detection of population level consequences of polyandry in elasmobranchs however–most of the species included in this study are subject to intense anthropogenic fishing pressure [24, 50], a relatively new selective pressure evolutionarily speaking that would undoubtedly provide the environmental change necessary to detect demographic effects. Even so, we failed to find any evidence of population-level consequences of genetic mating system variation (Table 3). Assuming the data used in this study are sufficient to detect hypothetical population-level consequences (see below for caveats associated

**Table 2. AIC values for models of trait evolution fit to MP frequency data.**

| Model of trait evolution | AIC | ΔAIC |
|---|---|---|
| Brownian motion (BM) | 137.8 | - |
| Ornstein Uhlenbeck (OU) | 117.2 | -20.6 |
| Early burst (EB) | 139.8 | 2.00 |

**Table 3. Output of phylogenetic linear models demonstrates the absence of any significant relationships between MP frequency and potential covariates.** Note that AIC values should not be used to compare directly between these models as they differ in sample size.

| Covariate | n | Gradient | α | σ² | P value | AIC |
|---|---|---|---|---|---|---|
| Conservation status | 41 | -2.92 | 0.01 | 20.68 | 0.46 | 392.9 |
| Lifetime reproductive output | 29 | -0.12 | 0.06 | 0.08 | 0.55 | 89.45 |
| Generation time | 19 | -0.17 | 0.06 | 0.10 | 0.48 | 61.82 |
| Minimum litter size | 35 | 0.29 | 0.01 | 0.02 | 0.12 | 102.4 |
| Maximum litter size | 35 | 0.17 | 8.77e-03 | 0.01 | 0.30 | 103.9 |
| Fecundity | 35 | 0.15 | 0.03 | 0.06 | 0.39 | 105.7 |
| Resilience | 41 | 0.25 | 9.28e-03 | 0.01 | 0.08 | 116.0 |
| Fishing vulnerability | 41 | 0.24 | 0.01 | 0.02 | 0.13 | 116.3 |
| Climate vulnerability | 19 | 0.16 | 0.19 | 3.54e-06 | 0.51 | 62.39 |

with this), this leaves two potential explanations: either mating system variation in elasmobranchs genuinely has no noticeable effect on population status or extinction risk, or there appears to be no noticeable effect due to costs and benefits of polyandry that are approximately equal in magnitude. The first of these scenarios is unlikely–there is evidence to suggest that genetic monogamy in sharks may reduce genetic variation and effective population sizes [22, 51, 52], and negative fitness consequences of unwanted mating attempts have been documented in multiple species [20, 53, 54]. Reduced genetic variation is likely of great significance to elasmobranch taxa, which exhibit slower rates of molecular evolution than other vertebrates [55, 56]. Thus, we suggest that mongogamy/polyandry in sharks may not exhibit population-level consequences as both endpoints of the mating system spectrum convey costs and benefits that are broadly equivalent in terms of their impacts on extinction risk.

## The evolution of mating system variation in elasmobranchs

Whilst we did not directly test any of the hypothesised adaptive explanations for mating system variation in elasmobranchs, our results do provide insight into the evolutionary dynamics underlying macroevolutionary shifts in MP frequency. Only one previous study has investigated polyandry in elasmobranchs in a phylogenetic context, finding that polyandry was likely ancestral to the clade [4]. Whilst this was an important initial step, the approach taken by Lamarca et al. [4] was limited by the use of a binary trait to describe the presence or absence of polyandry. This clearly represents an oversimplification of the true diversity of genetic mating systems observed in elasmobranchs, as evidenced by the variation in MP frequency in extant species (Table 1). Instead using MP frequency (a continuous variable, more representative of variation in genetic mating systems) we did not find statistically robust evidence for polyandry representing the ancestral state in elasmobranchs. This does not imply that genetic monogamy was the ancestral state, but simply that existing data are insufficient to provide robust estimation of the ancestral genetic mating system in elasmobranchs. Despite this uncertainty, there is clear evidence that both very high (>80%) and very low (<20%) levels of polyandry have evolved multiple times independently (Fig 1). This leads us to suggest that both genetic monogamy and polyandry represents alternative adaptive mating systems in Elasmobranchii (as previously suggested for other clades, see [14], each of which may be favoured under different biological and ecological conditions. Our finding that the phylogenetic distribution of MP frequency is best explained by an Ornstein-Uhlenbeck model of trait evolution (Table 2) provides further support for this, implying that MP frequency in elasmobranchs is evolving towards at least one adaptive peak [57]. As for what selective pressures could be underlying

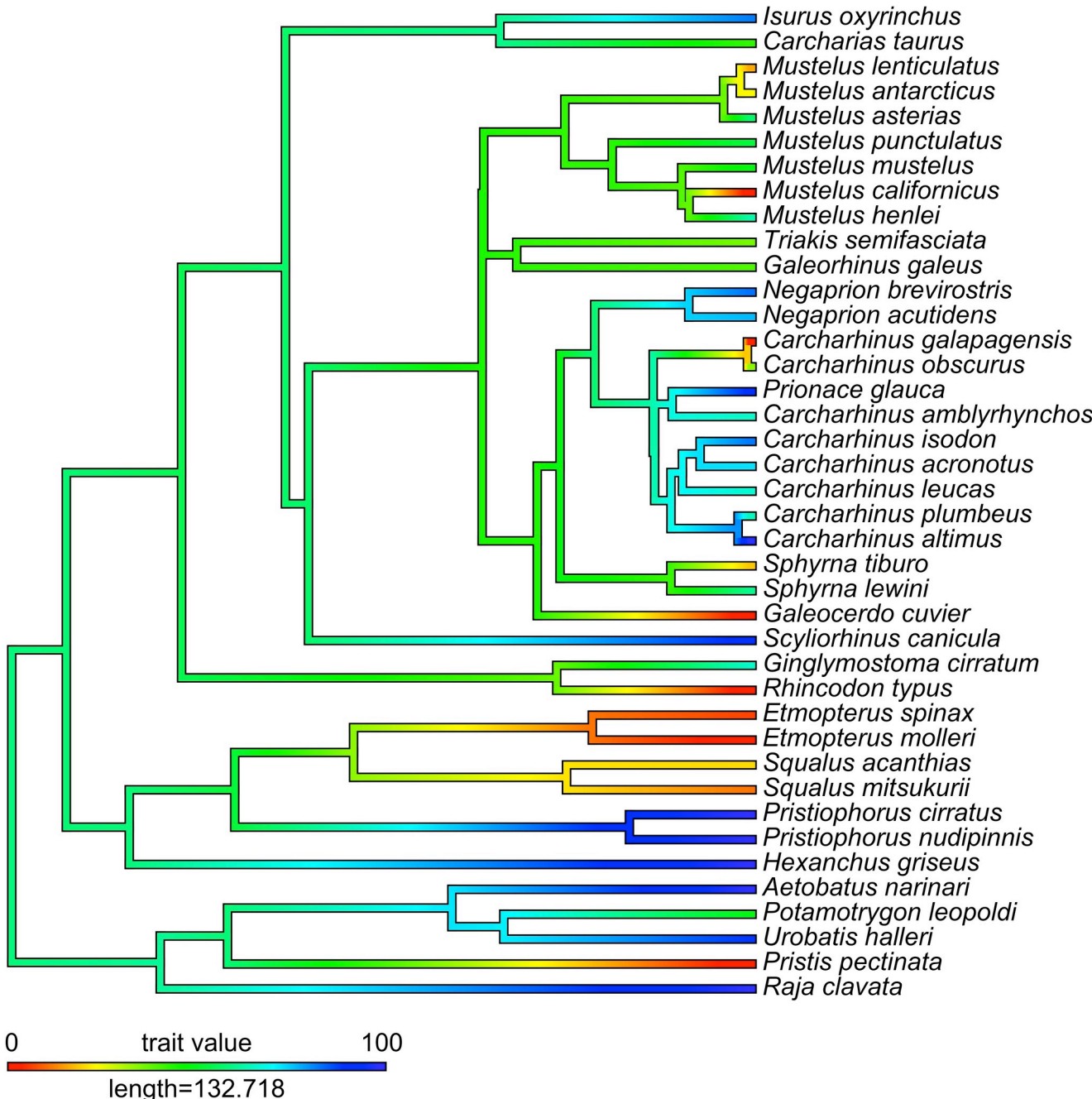

**Fig 1. Ancestral state reconstruction of MP frequency through elasmobranch phylogeny.** Red coloration represents genetic monogamy (low values of MP frequency), blue coloration represents polyandry (high MP frequency values) and green coloration represents intermediate MP frequency values.

these adaptive peaks, further work is needed, as consensus is yet to be reached regarding the adaptive basis of either genetic monogamy or polyandry in elasmobranchs [4, 21, 22].

## Limitations

We recognise that there are several fundamental limitations underlying the analyses presented in this study, and our understanding of mating system variation in elasmobranchs more

broadly. The need for increased taxonomic coverage of mating system studies in elasmobranchs is clear, with several major clades entirely data deficient [4] and the potential for alternative mating strategies such as polygyny entirely overlooked. This paucity of data not only restricts our understanding of the mating system variation in extant species, but reduces the robustness of comparative phylogenetic analyses such as ancestral state reconstruction. Most pertinent to the topic of this study, we are unable to categorically rule out the possibility of population-level consequences of polyandry or monogamy despite finding no evidence of such consequences (Table 3). We incorporated only one set of data per species, when in reality there is substantial regional variation in both reproductive biology and population status in many elasmobranch species [29, 58, 59]. Furthermore, differences in average population status between species are driven by a number of additional factors including life history traits and applied fishing pressure [60, 61], meaning that even if relationships between mating system and extinction risk existed, they may be cryptic and difficult to detect. Nonetheless, we find that on the basis of current data there is no substantial evidence for such relationships (Table 3). Thus, even if cryptic relationships between mating system and population status do exist, they are unlikely to be significant determinants of extinction vulnerability in extant elasmobranch species.

## Conclusions

Polyandry and monogamy clearly both have important consequences for individual fitness, in the case of both males and females [7, 62]. Despite theoretical predictions, it remains unclear whether mating system variation has substantial population-level effects however [10]. In Elasmobranchii, a large vertebrate radiation exhibiting substantial variation in genetic mating system, there is no evidence for such effects (Table 3). Both genetic monogamy and polyandry have evolved multiple times independently from an ancestor of unknown mating system, and there is evidence that the variation observed in extant elasmobranch species is adaptive (Fig 1 and Table 2). Nevertheless, the specific conditions under which genetic monogamy and polyandry are favoured remain uncertain and may be taxon specific.

## Supporting information

**S1 Table. Data underlying the analyses and results of this article.**
(CSV)

## Author Contributions

**Conceptualization:** Joel H. Gayford.

**Data curation:** Joel H. Gayford, Estefanía M. Flores-Flores.

**Formal analysis:** Joel H. Gayford.

**Investigation:** Joel H. Gayford.

**Methodology:** Joel H. Gayford.

**Writing – original draft:** Joel H. Gayford, Estefanía M. Flores-Flores.

**Writing – review & editing:** Joel H. Gayford, Estefanía M. Flores-Flores.

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
