## [Decision Letter · Decision Letter 0]

11 Jun 2024

PONE-D-24-08235No evidence for population-level benefits of polyandry in sharks and raysPLOS ONE

Dear Dr. Gayford,

Thank you for submitting your manuscript to PLOS ONE. After careful consideration, we feel that it has merit but does not fully meet PLOS ONE’s publication criteria as it currently stands. Therefore, we invite you to submit a revised version of the manuscript that addresses the points raised during the review process. It was extremely difficult to find reviewers for this manuscript. Finally I received very constructive comments. The paper needs major revision but has potential to be accepted after carefuly consideration of the reviewers concerns. 

We look forward to receiving your revised manuscript.

Kind regards,

Stefan Schlatt

Academic Editor

PLOS ONE

Journal Requirements:

Additional Editor Comments:

The authors have compiled an interesting study on the role of polyandry in sharks and rays. I agree with both reviewers that the study is of value but has also significant limitations. Both reviewers make constructive comments and recommendations for revision. I am optimistic that after point-by-point review and appropriate revision of the manuscript, it will be a useful reference and acceptable for publication.

Reviewers' comments:

Reviewer's Responses to Questions

**Comments to the Author**

1. Is the manuscript technically sound, and do the data support the conclusions?

Reviewer #1: Partly

Reviewer #2: Partly

2. Has the statistical analysis been performed appropriately and rigorously? 

Reviewer #1: Yes

Reviewer #2: No

3. Have the authors made all data underlying the findings in their manuscript fully available?

Reviewer #1: Yes

Reviewer #2: Yes

4. Is the manuscript presented in an intelligible fashion and written in standard English?

Reviewer #1: Yes

Reviewer #2: Yes

5. Review Comments to the Author

Reviewer #1: The authors present an MS where they evaluated the possible benefits of polyandry at the population level in elasmobranchs. In general, it is an interesting article, however, currently, there is a consensus that a mating system is not necessarily better than others and the costs and benefits they offer are circumstantial. I mentioned this because the authors seem to part from this idea. I recommend reading some articles on this matter: Karl 2008 (DOI 10.1111/j.1365-294X.2008.03902.x), Lotterhos 2011 (DOI 10.1111/j.1558-5646.2011.01249.x), and Stephens & Sutherland 2000 (Chapter 9 in the book Vertebrate Mating Systems).

I think that the methodology to abord this idea is adequate, in this sense, the article could give more emphasis to the multi-model approach they use to put this idea to test.

What I do consider worth exploring in the discussion in more depth is how the benefits of polyandry and monogamy together can positively influence populations.

To help the authors, I will try to give comments throughout the document in such a way that it can be useful to improve the MS.

Abstract

Lines 27-28. In this sentence, authors should specify this in the elasmobranch group, I suppose that’s what they mean.

Line 30. This is plural, those are evidence to each effect.

Lines 32-35. This sentence is so complex, that it has a justification and objective, this became too confusing for an abstract. I suggest restructuring in two sentences.

Lines 37-41. I suggest restructuring this sentence because it sounds contradictory to the objective in the abstract. Also, it sounds a little catastrophic when you said We fail! I recommend this…. Results showed that both genetic monogamy and polyandry have evolved multiple times independently….and in biological conditions.

Lines 39-41 and 42-43. I think that this idea has been discussed in some articles. Karl in 2008 debated this idea and he demonstrated that monogamy is not always a maladaptive strategy. For its part, Lotterhos 2011 also refuted this idea with a model, which showed that any adverse effect derived from polyandry is attenuated by long generation times. Kvarnemo's 2018 observations about monogamy suggest that this cannot be a maladaptive strategy since it is present in many groups, even if its benefits are not in sight.

Introduction

Line 56. “rampant polyandry” how can I interpret it?

Line 62. Add in parentheses, revised in 10. Because it is a single cite of information in this sentence and it is controversial.

Line 65. I have a conflict with these cites (10-11). Although one of them is proposed for the animal kingdom, in general, it is necessary to think about the peculiarities of the elasmobranch group. My main conflict is in the citation of Simmons 2001 since he bases his arguments on insects. Please, you need to consider this for vertebrates and, particularly in elasmobranchs.

Line 85. What do you mean by “unwanted mating attempts”?

Lines 91-94. You need to cite this sentence.

Lines 101-103. This sentence reflects that there are many mating systems and not just polyandry and monogamy, what would those be? The authors assume that readers understand the limitations of mating system studies in elasmobranchs, it seems to me not necessarily. By this, I mean that polygyny is very feasible in this group, although it is extremely difficult to verify.

Methodology

Lines 120-121. I agree that there is no best form to get a value of MP frequency for a species, however, it cannot reflect the actual mating system in species that have geographical variations, or seasonal variation. Also, the sample size is an important factor influencing the MP frequency in different studies for the same species, along with the resolution power offered by the different molecular tools employed. I suggest adding this in the limitations section, or better, stating those understandable limitations right after the methods.

Lines 123-125. Assuming this, authors need to discuss it in species with geographical variation in the MP.

Lines 131-133. I invite to authors to read the articles before mentioned. I consider that it is not reflected in the introduction.

Results

Table 2. This table is unnecessary. I suggest sending it to supplementary materials and citing it in this part.

Lines 187-189. I’m not familiar with the analysis, however, looking at the tree, I think the analyses give a lot of weight to the MP value. I’m concerned that in some cases, such as M. californicus, the sample size is too low to consider real genetic monogamy, since a particularity of this work is possibly collecting small females with a low probability of observing MP. In this case, It can be an effect of the sample size.

Discussion

Lines 206-209. These ideas should be expressed in the abstract and of course in the introduction.

Lines 220-222. I’m not sure if this is correct according to the abstract of Dibattista et al. 2008. In this abstract, they said…” We find that offspring from polyandrous litters did not have a greater genetic diversity or greater survival than did the offspring of monandrous litters”. This sentence is contrary evidence.

Lines 225-228. This idea is correct and is the best of all paper. This idea has been discussed. I suggest to the authors redirect the justification of their article considering their methodology.

Lines 239-241. Using the average MP in a species cannot reflect the real mating system.

Lines 246-248. Please, read to Stephens & Sutherland 2000, they had already discussed it and explained the Allee effect in this context.

Lines 248-251. I do not agree. Since the model was not significant, I think the recurrent MP is not necessarily one adaptive peak. In this sense, this sentence is contradictory to the last sentence. Also, we need to consider that there are few studies to evaluate the mating system in the Elasmobranchii clade, thus this apparently adaptive peak can be skewed due to limited publications.

Limitations

Lines 264-266. Excellent! Authors need to moderate some previous sentences according to this section.

Conclusions

Lines 282-283. I think this is not demonstrated with your results.

Reviewer #2: The potential link between multiple paternity, sexual selection and population viability is an important and general one, and I agree with the authors that this taxonomic group is a potentially useful model system in which to study it. That being said, I found it a little difficult to know what to conclude from this study. In principle I think it is important that negative findings are published, but I do have concerns as to whether the study really had sufficient power to test the hypothesis in question. To be fair to the authors, they are quite open about the limitations of the dataset in their discussion, so on balance I think I would be in favour, but first would like to see several more minor points addressed:

Line 117: The data are presented as coming form the existing literature, but ca. 80% come from a single source. For that reason, I think it would be appropriate to present already in the Introduction the potential overlaps and novel aspects relative to this previous study.

143: It’s unclear to me why conservation status was coded quantitatively, rather than adopt a different analysis approach for this variable? Linear regression doesn’t feel like the most appropriate approach here.

173-177: Given all of these “relationships” are non-significant, I don’t think it is very meaningful to discuss which slopes are positive and which are negative; all you can reliably conclude is that none could be distinguished from zero.

200: on the face of it, purging would tend to reduce genetic diversity rather than increasing it - maybe take a bit more space to explain this point?

219: I understand you wished to use these investigated variables as proxies for extinction risk, but given you only look at extant species in this analysis I think this might be slightly overstated here.

244: the argument that both very high and very low polyandry evolves frequently in this group seems to support the approach of Lamarca et al.? I do think there is a real risk that using only one estimate of MP per species gives a falsely precise impression, since this presumably varies a lot from population to population. To be fair, this point is acknowledged later in the discussion, but is there scope to quantify this to give an idea about the scale of the problem? for how many species did you average across multiple populations?

Table 3: please provide per analysis sample sizes, rather than just saying these varied. I also saw no sign that you accounted for multiple hypothesis testing here - these are essentially 9 different tests of the same underlying hypothesis, so there ought to be some control of the global false positive rate. Given all 9 are non-significant this doesn’t really matter in practice, but in principle it’s an issue that could at least be acknowledged.

A few small typos/errors to correct:

36: through not though

56: for me rampant has potentially negative connotations, consider replacing with something more neutral (widespread?)

88: cost *of* mating

113: consequently

118: reference formatting

130: reference formatting

153: series *of*

259: missing word?

6. PLOS authors have the option to publish the peer review history of their article (what does this mean?). If published, this will include your full peer review and any attached files.

Reviewer #1: **Yes: **Nancy C. Saavedra-Sotelo

Reviewer #2: No

---

## [Author Response · Author response to Decision Letter 0]

11 Jun 2024

We thank the editor and reviewer for their time spent on this manuscript. We are very grateful for both of your comments, and believe that they have greatly improved the quality of the paper. Please find detailed responses to comments below. We hope that you find the changes made sufficient to consider acceptance of the manuscript, as we have to the best of our abilities incorporated all changes suggested by the reviewer into the text.

Reviewer 1

Comment: The authors present an MS where they evaluated the possible benefits of polyandry at the population level in elasmobranchs. In general, it is an interesting article, however, currently, there is a consensus that a mating system is not necessarily better than others and the costs and benefits they offer are circumstantial. I mentioned this because the authors seem to part from this idea. I recommend reading some articles on this matter: Karl 2008 (DOI 10.1111/j.1365-294X.2008.03902.x), Lotterhos 2011 (DOI 10.1111/j.1558-5646.2011.01249.x), and Stephens & Sutherland 2000 (Chapter 9 in the book Vertebrate Mating Systems).

I think that the methodology to abord this idea is adequate, in this sense, the article could give more emphasis to the multi-model approach they use to put this idea to test.

What I do consider worth exploring in the discussion in more depth is how the benefits of polyandry and monogamy together can positively influence populations.

Response: Thank you for your comment. We have read the papers you suggest and cited them where relevant in the manuscript.

Comment: Lines 27-28. In this sentence, authors should specify this in the elasmobranch group, I suppose that’s what they mean.

Response: Thank you for your comment. This sentence is general, and is referring to the entire animal kingdom. Several papers (including reference 10 in our list) explain how the theoretical consequences of mating system variation are applicable across animal life, and that the biological contexts that could result in different benefits or costs to polyandry or monogamy can occur independent of phylogeny. 

Comment: Line 30. This is plural, those are evidence to each effect.

Response: Thank you for your comment. The word ‘these’ used in the original manuscript is plural and correct in this context and thus we have decided to keep it.

Comment: Lines 32-35. This sentence is so complex, that it has a justification and objective, this became too confusing for an abstract. I suggest restructuring in two sentences.

Response: Thank you for your comment. The requested change has been made.

Comment: Lines 37-41. I suggest restructuring this sentence because it sounds contradictory to the objective in the abstract. Also, it sounds a little catastrophic when you said We fail! I recommend this…. Results showed that both genetic monogamy and polyandry have evolved multiple times independently….and in biological conditions.

Response: Thank you for your comment. The requested change has been made.

Comment: Lines 39-41 and 42-43. I think that this idea has been discussed in some articles. Karl in 2008 debated this idea and he demonstrated that monogamy is not always a maladaptive strategy. For its part, Lotterhos 2011 also refuted this idea with a model, which showed that any adverse effect derived from polyandry is attenuated by long generation times. Kvarnemo's 2018 observations about monogamy suggest that this cannot be a maladaptive strategy since it is present in many groups, even if its benefits are not in sight.

Response: Thank you for your comment. We have added additional text in lines 81-86 and cited the requested references.

Comment: Line 56. “rampant polyandry” how can I interpret it?

Response: Thank you for your response. We have replaced the word rampant with widespread.

Comment: Line 62. Add in parentheses, revised in 10. Because it is a single cite of information in this sentence and it is controversial.

Response: Thank you for your comment. The requested change has been made.

Comment: Line 65. I have a conflict with these cites (10-11). Although one of them is proposed for the animal kingdom, in general, it is necessary to think about the peculiarities of the elasmobranch group. My main conflict is in the citation of Simmons 2001 since he bases his arguments on insects. Please, you need to consider this for vertebrates and, particularly in elasmobranchs.

Response: Thank you for your comment. For the purpose of this sentence, we do not see that any specific reference to elasmobranchs is required. Our understanding of the net fitness consequences of polyandry are poor across the animal kingdom, and the specific sentence you query was not talking only about elasmobranchs, but about our understanding of polyandry across animal life. The studies we cite are some of the only empirical tests of hypotheses concerning net fitness consequences, so when making a general statement such as “understanding of the net fitness consequences of polyandry are poor across the animal kingdom” they provide ideal references. We do go on to discuss later specific knowledge regarding elasmobranchs, but we do not believe that there is any issue with including these studies given the general nature of the statement.

Comment: Line 85. What do you mean by “unwanted mating attempts”?

Response: Thank you for your comment. We have clarified this in the text by replacing unwanted mating attempts with coercive mating

Comment: Lines 91-94. You need to cite this sentence.

Response: Thank you for your comment. We have made the change as requested.

Comment: Lines 101-103. This sentence reflects that there are many mating systems and not just polyandry and monogamy, what would those be? The authors assume that readers understand the limitations of mating system studies in elasmobranchs, it seems to me not necessarily. By this, I mean that polygyny is very feasible in this group, although it is extremely difficult to verify.

Response: Thank you for your comment. We have included additional text in lines 129-130 to clarify for the reader. We also provide additional text mentioning polygyny in lines 348-349

Comment: Lines 120-121. I agree that there is no best form to get a value of MP frequency for a species, however, it cannot reflect the actual mating system in species that have geographical variations, or seasonal variation. Also, the sample size is an important factor influencing the MP frequency in different studies for the same species, along with the resolution power offered by the different molecular tools employed. I suggest adding this in the limitations section, or better, stating those understandable limitations right after the methods.

Response: Thank you for your comment. We have addressed this issue in the methodology in lines 149-158.

Comment: Lines 123-125. Assuming this, authors need to discuss it in species with geographical variation in the MP.

Response: Thank you for your comment. We have added extra text to acknowledge this limitation and justify our approach in the methodology, in lines 149-158

Comment: Lines 131-133. I invite to authors to read the articles before mentioned. I consider that it is not reflected in the introduction.

Response: Thank you for your comment. As mentioned in our response to your previous comment we have read the articles and added text to the introduction as requested.

Comment: Table 2. This table is unnecessary. I suggest sending it to supplementary materials and citing it in this part.

Response: Thank you for your comment. We have decided to keep this table as it reports an important result that is referenced throughout the discussion. 

Comment: Lines 187-189. I’m not familiar with the analysis, however, looking at the tree, I think the analyses give a lot of weight to the MP value. I’m concerned that in some cases, such as M. californicus, the sample size is too low to consider real genetic monogamy, since a particularity of this work is possibly collecting small females with a low probability of observing MP. In this case, It can be an effect of the sample size.

Response: Thank you for your comment. This is true to some extent, however the issues affecting this analyses are the same as those affecting the other analyses, and thus have been addressed in the limitations section. Even excluding cases with very low sample size there are still multiple cases in which very low values of MP have evolved, and even in species with low sample size it is unlikely that additional studies would find values dramatically different from existing studies. That is the main benefit of using a continuous measure of MP as opposed to a binary variable, as it allows use of MP as an approximation rather than a discrete variable that must show either monogamy or polyandry. We have added additional text in lines 149-158 to explain this.

Comment: Lines 206-209. These ideas should be expressed in the abstract and of course in the introduction.

Response: Thank you for your comment. This idea is already mentioned in lines 30-32 of the original abstract. We have added additional text in lines 75-79 in the introduction to further expand.

Comment: Lines 220-222. I’m not sure if this is correct according to the abstract of Dibattista et al. 2008. In this abstract, they said…” We find that offspring from polyandrous litters did not have a greater genetic diversity or greater survival than did the offspring of monandrous litters”. This sentence is contrary evidence.

Response: Thank you for your comment. DiBattista et al. 2008 do not present any data regarding monogamy, so their findings are not relevant to the sentence. However in the main text they do state that monogamy in sharks may reduce genetic variation, and hence this is why we included the reference. 

Comment: Lines 225-228. This idea is correct and is the best of all paper. This idea has been discussed. I suggest to the authors redirect the justification of their article considering their methodology.

Response: Thank you for your comment. We believe that framing the paper in terms of the empirical results is the most scientifically robust choice. If you disagree with this we would welcome additional explanation of how you would alter the article’s focus or justification.

Comment: Lines 239-241. Using the average MP in a species cannot reflect the real mating system.

Response: Thank you for your comment. We believe that when comparing across species the approximation of average MP is the best available measure, as explained in the new text additions to the methodology. However as described in our response to a previous comment we have added text to the methodology to acknowledge this limitation.

Comment: Lines 246-248. Please, read to Stephens & Sutherland 2000, they had already discussed it and explained the Allee effect in this context.

Response: Thank you for your comment. We have read and cited the reference as requested.

Comment: Lines 248-251. I do not agree. Since the model was not significant, I think the recurrent MP is not necessarily one adaptive peak. In this sense, this sentence is contradictory to the last sentence. Also, we need to consider that there are few studies to evaluate the mating system in the Elasmobranchii clade, thus this apparently adaptive peak can be skewed due to limited publications.

Response: Thank you for your comment. We believe you may have been looking at the wrong table. Table 2 shows the results of our evolutionary model test. These tests do not come with p values and hence do not have a strict ‘significance’. They are compared on the basis of AIC, where a difference of 2 or more is considered significant. The Ornstein-Uhlenbeck model received substantially more support than the null model of Brownian motion. Therefore our results suggest that MP in elasmobranchs follows an OU model of trait evolution, in which there are one or more adaptive peaks. Of course the paucity of data does affect this analysis like all of the other analyses, and the limitations section was included specifically to address this.

Comment: Lines 264-266. Excellent! Authors need to moderate some previous sentences according to this section.

Response: Thank you for your comment. See our response to other comments for details of how we have made the requested changes.

Comment: Lines 282-283. I think this is not demonstrated with your results.

Response: Please see the above comment regarding Table 2. We found substantial support for an OU model of trait evolution over the null model of Brownian motion, which implies that MP is evolving towards one or more adaptive peaks.

Reviewer 2

Comment: The potential link between multiple paternity, sexual selection and population viability is an important and general one, and I agree with the authors that this taxonomic group is a potentially useful model system in which to study it. That being said, I found it a little difficult to know what to conclude from this study. In principle I think it is important that negative findings are published, but I do have concerns as to whether the study really had sufficient power to test the hypothesis in question. To be fair to the authors, they are quite open about the limitations of the dataset in their discussion, so on balance I think I would be in favour, but first would like to see several more minor points addressed:

Response: Thank you for your comment. We agree with the sentiment you convey. Of course the necessary data to robustly prove the fitness consequences and population level consequences of mate system variation in elasmobranchs do not yet exist. However, we believe this study is valuable as it not only provides valuable insight into the evolution of mating system variation in elasmobranchs, but may serve as the impetus for future studies that build on the existing data. We have endeavoured to make the requested changes wherever possible and hope that the revised manuscript addresses the concerns raised below.

Comment: Line 117: The data are presented as coming form the existing literature, but ca. 80% come from a single source. For that reason, I think it would be appropriate to present already in the Introduction the potential overlaps and novel aspects relative to this previous study.

Response: Thank you for your comment. The cited paper did itself gather these values from existing literature. The novelty of our study relative to Lamarca et al is discussed in lines 92-97. Essentially, their analyses is very simplistic, and doesn’t at all address potential population-level consequences. Rather it is entirely focussed on addressing the ancestral state of genetic mating system in elasmobranchs, however their analyses is very basic in this regard compared to our ancestral state reconstruction (Figure 3). 

Comment: 143: It’s unclear to me why conservation status was coded quantitatively, rather than adopt a different analysis approach for this variable? Linear regression doesn’t feel like the most appropriate approach here.

Response: Thank you for your comment. After reviewing the approach we used, we have modified the text slightly in the methodology to explain better. The models we fit were phylogenetic generalised least squares regression models, which is a form of linear regression that takes into account phylogenetic non-independence. These models can incorporate discrete and continuous variables, and therefore using conservation status is entirely valid. We removed the text referencing coding as this has no effect on the results.

Comment: 173-177: Given all of these “relationships” are non-significant, I don’t think it is very meaningful to discuss which slopes are positive and which are negative; all you can reliably conclude is that none could be distinguished from zero.

Response: Thank you for your comment. We have now deleted this portion of the text.

Comment: 200: on the face of it, purging would tend to reduce genetic diversity rather than increasing it - maybe take a bit more space to explain this point?

Response: Thank you for your comment. Indeed the references to purging and genetic variation were supposed to be separate, and as such we have altered the sentence order to explain that these are 

---

## [Decision Letter · Decision Letter 1]

3 Jul 2024

PONE-D-24-08235R1No evidence for population-level benefits of polyandry in sharks and raysPLOS ONE

Dear Dr. Gayford,

Thank you for submitting your manuscript to PLOS ONE. After careful consideration, we feel that it has merit but does not fully meet PLOS ONE’s publication criteria as it currently stands. Therefore, we invite you to submit a revised version of the manuscript that addresses the points raised during the review process. The revision of this paper has been a long journey. At this point the manuscript is almost appropriately revised and will not go back into re-review. However the reviewer mentions a few minor changes which I like to ask the authors to consider and implement. The paper will then be acceptable.

We look forward to receiving your revised manuscript.

Kind regards,

Stefan Schlatt

Academic Editor

PLOS ONE

Journal Requirements:

Reviewers' comments:

Reviewer's Responses to Questions

**Comments to the Author**

1. If the authors have adequately addressed your comments raised in a previous round of review and you feel that this manuscript is now acceptable for publication, you may indicate that here to bypass the “Comments to the Author” section, enter your conflict of interest statement in the “Confidential to Editor” section, and submit your "Accept" recommendation.

Reviewer #1: (No Response)

2. Is the manuscript technically sound, and do the data support the conclusions?

Reviewer #1: Partly

3. Has the statistical analysis been performed appropriately and rigorously? 

Reviewer #1: Yes

4. Have the authors made all data underlying the findings in their manuscript fully available?

Reviewer #1: Yes

5. Is the manuscript presented in an intelligible fashion and written in standard English?

Reviewer #1: Yes

6. Review Comments to the Author

Reviewer #1: I consider that the authors aboarded all my comments and questions, especially in the interpretation of results.

I agree with almost all editions that they made in the manuscript. However, I have some comments that I would like to see in the paper. In this sense, I suggest another round of revision.

Abstract

Lines 27-28. Please specify if this idea is for the elasmobranch clade or animal kingdom.

Lines 37-40. I feel that the authors handle this idea a lot, but the topic is not delved into in the discussion, they leave everything to the reader's imagination. In this sense, I have a comment on the discussion.

Lines 43-44. I insist that this idea is no longer under discussion currently, even in elasmobranchs. Also, it doesn't delve into it, I suggest eliminating it.

Discussion

Lines 239-242. I suggest exploring this idea a little further in the discussion, I can imagine what they are referring to, but I think it would be better to read it. Since the article is titled "No evidence for population-level benefits of polyandry in sharks and rays" I hope to see some examples in this regard to understand much better how the cancellation of both characteristics (benefits and costs) of the mating system does not necessarily impact the risk of extinction of a species.

7. PLOS authors have the option to publish the peer review history of their article (what does this mean?). If published, this will include your full peer review and any attached files.

Reviewer #1: No

---

## [Author Response · Author response to Decision Letter 1]

4 Jul 2024

Response to reviewers: No evidence for population-level benefits of polyandry in sharks and rays

We thank the editor and reviewer for their time spent on this manuscript. We are very grateful for both of your comments, and believe that they have greatly improved the quality of the paper. Please find detailed responses to comments below. We hope that you find the changes made sufficient to consider acceptance of the manuscript, as we have to the best of our abilities incorporated all changes suggested by the reviewer into the text.

Reviewer 1

Comment: Lines 27-28. Please specify if this idea is for the elasmobranch clade or animal kingdom.

Response: The sentence has been clarified as requested.

Comment: Lines 37-40. I feel that the authors handle this idea a lot, but the topic is not delved into in the discussion, they leave everything to the reader's imagination. In this sense, I have a comment on the discussion.

Response: Please see our response to the comment about the discussion.

Comment: Lines 43-44. I insist that this idea is no longer under discussion currently, even in elasmobranchs. Also, it doesn't delve into it, I suggest eliminating it.

Response: The statement has been eliminated as requested.

Comment: Lines 239-242. I suggest exploring this idea a little further in the discussion, I can imagine what they are referring to, but I think it would be better to read it. Since the article is titled "No evidence for population-level benefits of polyandry in sharks and rays" I hope to see some examples in this regard to understand much better how the cancellation of both characteristics (benefits and costs) of the mating system does not necessarily impact the risk of extinction of a species.

Response: The current version of the manuscript contains quite a lot of discussion of the costs and benefits of different mating systems. See lines 62-80 and lines 243-255. The fact is that all of these costs and benefits are highly context dependent (as we point out at several points in the text), so we cannot make detailed inferences about specific costs and benefits in specific shark or ray species as we lack the necessary contextual information. The current sections on costs and benefits address to the fullest extent possible how each of these factors could apply to sharks and rays, and provide references that delve deeper into the theoretical basis of those costs and benefits, should the reader be interested.

---

## [Editor Report · Decision Letter 2]

9 Jul 2024

PONE-D-24-08235R2No evidence for population-level benefits of polyandry in sharks and raysPLOS ONE

Dear Dr. Gayford,

Thank you for submitting your manuscript to PLOS ONE. After careful consideration, we feel that it has merit but does not fully meet PLOS ONE’s publication criteria as it currently stands. Therefore, we invite you to submit a revised version of the manuscript that addresses the points raised during the review process.

I am happy to see that the paper has been appropriately revised. One reviewer expresses minor concerns which I like to see addressed before the paper can be accepted for publication.

We look forward to receiving your revised manuscript.

Kind regards,

Stefan Schlatt

Academic Editor

PLOS ONE
---

## [Author Response · Author response to Decision Letter 2]

9 Jul 2024

The reference list has been checked, and is fully complete. No additional references need to be removed or added to the list. We have made some minor formatting adjustments for consistency as can be seen in the tracked changes document.

---

## [Editor Report · Decision Letter 3]

18 Jul 2024

No evidence for population-level benefits of polyandry in sharks and rays

PONE-D-24-08235R3

Dear Dr. Gayford,

We’re pleased to inform you that your manuscript has been judged scientifically suitable for publication and will be formally accepted for publication once it meets all outstanding technical requirements.

Kind regards,

Stefan Schlatt

Academic Editor

PLOS ONE
---

## [Editor Report · Acceptance letter]

19 Jul 2024

PONE-D-24-08235R3 

PLOS ONE

Dear Dr. Gayford, 

I'm pleased to inform you that your manuscript has been deemed suitable for publication in PLOS ONE. Congratulations! Your manuscript is now being handed over to our production team.

Kind regards, 

on behalf of

Dr. Stefan Schlatt 

Academic Editor

PLOS ONE